# ‘*Be the Match*’. Predictors of Decisions Concerning Registration as a Potential Bone Marrow Donor—A Psycho-Socio-Demographic Study

**DOI:** 10.3390/ijerph20115993

**Published:** 2023-05-29

**Authors:** Jacek Bogucki, Wioletta Tuszyńska-Bogucka

**Affiliations:** 1Department of Organic Chemistry, Faculty of Pharmacy, Medical University of Lublin, 20-059 Lublin, Poland; 2Department of Human Sciences, WSEI University, 20-209 Lublin, Poland

**Keywords:** psychological aspects of donation, psycho-oncology, willingness for donation, potential bone marrow donor, psycho-socio-demographic profile, machine learning

## Abstract

(1) Background: The study was aimed at a better understanding of the factors determining making a decision to become a potential bone marrow donor, in a Polish research sample; (2) Methods: The data was collected using a self-report questionnaire among persons who voluntarily participated in the study concerning donation, conducted on a sample of the Polish population via Internet. The study included 533 respondents (345 females and 188 males), aged 18–49. Relationships between the decision about registration as potential bone marrow donor and psycho-socio-demographic factors were estimated using the machine learning methods (binary logistic regression and classification & regression tree); (3) Results. The applied methods coherently emphasized the crucial role of personal experiences in making the decision about willingness for potential donation, f.e. familiarity with the potential donor. They also indicated religious issues and negative health state assessment as main decision-making destimulators; (4) Conclusions. The results of the study may contribute to an increase in the effectiveness of recruitment actions by more precise personalization of popularizing-recruitment actions addressed to the potential donors. It was found that selected machine learning methods are interesting set of analyses, increasing the prognostic accuracy and quality of the proposed model.

## 1. Introduction

Hematopoietic stem cell transplantations have become a standard medical procedure and are currently increasingly used more often in the treatment of both cancerous and non-cancerous diseases. For only about 25–30% of patients is a family donor successfully found, whereas for 40% of patients from the remaining group, an unrelated donor may be successfully matched. The probability of finding an unrelated matched donor increases with the number of volunteers included in the Donor Registries [1]. It is therefore very important to constantly expand their number by carrying out increasingly effective information-recruitment campaigns.

The list of diseases in which an allogeneic hematopoietic cell transplant may be considered is constantly expanding. Leukemia constitutes 25.8% of all cases of cancer in children, adolescents, and young adults under the age of 20. Leukemia is the second most common cause of death due to cancer among children, adolescents, and young adults aged under 20 years, which is 26.1% of all deaths due to cancer in this age group [2]. The multiplicity of cases of cancerous diseases of this type and the increasingly younger age of patients forces modern societies to accept the necessity of developing the idea of donation and recruitment of a suitable number of potential bone marrow donors (PBMDs), which may contribute to helping to save many patients [1]. In many cases, the application of a bone marrow (BM) transplant in hematological diseases, especially hematological malignancies, primary immunodeficiency, and some solid tumors, have become the treatment of choice [3,4,5,6,7,8]. The multiplicity of cases involving cancerous diseases of this type, and the increasingly younger age of patients, forces modern societies to accept the necessity for developing the idea of donation and recruitment of a suitable number of potential donors [1,9,10,11], and hence contributing to saving many patients. Among donations, one important type is the altruistic donation of a living person, with particular consideration of donation designed to benefit unrelated people (Living Anonymous Donation to a Stranger—LADS). It is important to continue to expand the group of people ready to make an act of donation, where it is essential to develop programs for the recruitment of the largest number of people to act as potential donor biorepositories.

The effectiveness in the recruitment of donors may be optimized by addressing the population segments with the greatest resource available and focusing on those who most frequently react positively. However, it is not just about determining the profile of people who have already registered. Knowledge is also needed concerning both the typical profile of a potential donor and about those candidates who ‘escape’ recruitment. Only when it is known in which groups the current actions are ineffective is it possible to address the campaigns properly.

Personal profile. Studies suggest that people who have decided to support LADS (including PBMD) are relatively young (i.e., under 40), better educated, less conservative, less religious, and have a more positive attitude towards science [12,13,14,15]. Among PBMD members, a higher percentage of females was noted [16,17,18]. Remaining in an intimate relationship occurred to be a factor that positively correlated with the decision to become a potential donor [12,17]. It was confirmed that blood donation, having an acquaintance or friend who already registered as a PBMD, having a relative or friend who needed a bone marrow transplant, or remaining in a partnership relationship are predictive factors for the decision to register as a PBMD [17,19]. The reports in the literature are scarce concerning the effects of assessing the health status on making the decision about the readiness to become a BM donor; however, it was found that a negative evaluation of the state of health blocked this decision [16].

Family and parental factors. Psychological and social family factors seem to have a potential relation with the decision to become a donor; however, family studies refer mainly to understanding of the predictive factors related to the family consent ratio (post-mortem donations from relatives) on donation and family barriers in obtaining the consent from the family for the potential organ donation. Only some structural family factors, such as possessing siblings, gender, living in a two-parent family, etc. [20,21], were tested as potential predictive factors of a positive decision about altruistic behaviors; however, no reports have been found concerning the family characteristics of potential donors with consideration of both the family structure and family relations, education of the parents, and their donation and pro-social activity. Nevertheless, in considering knowledge in social and educational psychology concerning the effect of intergenerational transmission of attitudes and values [22,23], it may be presumed that family history may be an important guideline in seeking the predictors related to a readiness for donation decisions.

It seems that there is a lack of analyses considering all the determinants of decision making explored to date, considered jointly and analyzed using the impact testing methods (e.g., analysis of regression). Therefore, in the present study, an attempt was undertaken to jointly consider factors that, to date, have been considered separately. In this study, an exploratory-predictive model was applied for the following reasons: (1) in the studies available for analysis, there was a lack of a model concerning the impact of the investigated variables that would be subject to verification by means of the available methods of statistical analysis; (2) machine learning methods allow to the greatest degree the identification in the population of the groups predestined to the occurrence of a specified event, whereas the LR and CART are complementary and constitute a methodologically tested suit in this type of research (decisions) [18].

Simultaneous consideration of all these variables is innovatory and may help in the determination of a strategy related to social marketing for the recruitment centers in future campaigns.

## 2. Materials and Methods

### 2.1. Setting and Sample

The data were collected using a self-report questionnaire among people who voluntarily participated in a psycho-socio-demographic study concerning donation, conducted on a sample of the Polish population. The questionnaire included typically formulated questions concerning socio-economic status (SES—age, gender, material standard, state of health, etc.), items related to family structure, declaration of religiosity, and participation in religious practices. An important area were questions concerning knowing the beneficiary of a donation (‘Among relatives or friends, I have some people who have received help in the form of a transplantation of tissues or organs’) or potential donors (‘I have a close person or a friend who is an honorary blood donor, registered as a potential bone marrow donor, or signed a statement on organ donation’). The wording was based on an analysis of the available study projects. Author-constructed questions were also added concerning donation and pro-social behaviors of the parents, e.g., ‘My mother/father is/was an honorary blood donor, registered as a potential bone marrow donor, or signed a statement on organ donation’; ‘My mother/father is/was involved in a social activity (affiliation to social organizations, volunteering, charity, etc.)’, as well as an own activity: ‘I am engaged in social activity (affiliation with social organizations, volunteering, charity, etc.)’.

Participants were recruited after responding to an online announcement posted on social media (FB, events, discussion groups, sharing, snowballing) (1 January 2020–30 June 2021). In this way, one of the recommended selection strategies for Internet research [24] was used, consisting of selecting the largest possible groups of volunteers, recruited through many different places on the Internet. The announcement contained a URL link to a questionnaire delivered via Google, which today is an accepted research standard because of their satisfactory stability [25,26]. Participants did not receive any remuneration for participating in the study. The only inclusion criterion for participation was age (18–50). The time for completing the form was not limited. The study included 533 respondents, with 345 (64.7%) females and 188 (35.3%) males, aged 18–49 (M = 24, SD = 4.71); 520 of the respondents were aged up to 40 (97.5%) and 13 respondents were aged 40–48 (2.5%) (Table 1). The disproportion between the respondent groups (PBMD < no-PBMD) is basically a favorable circumstance, considering the postulates for the investigation of mainly barriers, which means focusing on non-donors [27]. Minimal sample size was calculated by the Calculator for Logistic Regression, UCSF Clinical and Translational Science Institute [28].

### 2.2. Methodological Approach

This is a descriptive correlational study in which the registration as a PBMD is considered as a dependent variable and the psycho-socio-demographic factors as independent variables as well as significant predictors. The research metamodel was conceptualized before the start of the study, on the basis of the notion that it would incorporate the same type of information as the literature review.

The aim of the study was to identify the strongest predictors related to decisions about registration as a PBMD, with consideration of the following factors: (1) personal profile (gender, age, level and type of education, environment of origin, declaration of religiosity and participation in religious practices, intimate relationship and the level of satisfaction with this relationship, having a relative/acquaintance who registered as a PBMD or was a beneficiary of the medical donation procedure, pro-social activity (i.e., volunteering) and assessment of material and health status), (2) family profile (family structure, possessing siblings, relationships between parents), and (3) parental factors (level of education, pro-social activity, and donation behaviors of parents), using the method of machine learning data analysis in order to perform predictive modelling.

Machine learning (ML) is a class of AI techniques whose task is to collect and process data sets, on the basis of which certain solutions may be made/projected/anticipated (in our case, it was the detection of the main predictors of the decision to register as a PBMD). In machine learning, algorithms are trained to find patterns and correlations in large data sets, and then make the best decisions and forecasts based on this analysis. Machine learning algorithms are used in many different applications, such as approximating an unknown function based on samples, establishing functional relationships in data, or predicting trends. From the available ML techniques, we chose logistic regression (LR, which places variables on a graph, but they do not form a straight line, rather an S-shape) and classification and regression tree (CART, which can classify data on the basis of categorical and continuous variables). The combination of these types of ML produces a trained file. New data can be entered into it so that it recognizes patterns and formulates forecasts, or makes decisions relating to the type of recipients of the information regarding bone marrow donation.

The analysis was performed using a statistical software package, SPSS 25. Relationships between the decision about registration as a PBMD and psycho-socio-demographic factors were estimated using the models of LR and CART. This results from the assumption of seeking the maximum likelihood function, which would allow for the determination of the set of parameters with the greatest accuracy as well as prognostic importance [29]. The advantage of the adopted set may be confirmed complementarity of the regression models, LR and CART, in medical studies. LR focuses on variables of a relatively high statistical significance, whereas CART does not optimize the fit of the model to the data but sequentially divides the examined population into subgroups, on the basis of the best prognostic variable (regression models focus on variables of a relatively high statistical significance, whereas decision trees do not optimize the adjustment of the model to the data but sequentially divide the examined population into subgroups, on the basis of the best prognostic variables—this allows for the identification of subgroups in the population predestined to the occurrence of a specified event) [18]. Decision tree algorithms are commonly used in machine learning to acquire knowledge that is based on examples. The purpose of decision tree algorithms can be to create a plan or to solve a decision-making problem.

## 3. Results

### 3.1. Step 1. Logistic Regression Analysis

In order to determine which of the analyzed variables exerted a significant effect on the registration as a PBMD, logistic regression analysis was performed, using the backward elimination method. The analysis results showed that the final model (after 14 steps) was characterized by a good fit to the data, which was indicated by a Hosmer–Lemeshow test χ^2^(8) = 5.21; *p =* 0.735. The adopted model allows for the the overall correctness of classification at the level of 78.2%, with the correctness of classification of the respondents who did not register as a PBMD at 91.6%, whereas the correctness of classification of respondents who registered as a PBMD was 40.5%.

LR analysis (Table 2) indicated an increase in the odds ratio (OR) of the occurrence of the analyzed event (registration as a PBMD), and the following predictors had a significant effect: having a relative/friend registered as a PBMD, pro-social activity of the mother, engagement in volunteering, having a relative/friend who needed and received donation help, being born in a medium-sized town (up to 500,000 population) but not in a larger town or a capital city, and friendly relationships between parents. The factors that significantly decreased the probability of registration were the secondary school education level, participation in religious practices, negative evaluation of own state of health, and place of birth in a medium-sized town.

The analysis of the odds ratios demonstrated that the chance for registration as a PBMD among respondents with a secondary education level was 66% lower, compared to those with a higher education (OR = 0.34; 95% CI =0.17–0.67). In the case of respondents who assessed their health status as poor, the chance for registration as a PBMD was 92% lower than those who reported a good state of health (OR = 0.08; 95% CI = 0.01–0.53). The chance for registration as a PBMD was 5.15 times higher among respondents who evaluated the relationship between parents as friendly, compared to those who considered this relationship as poor (OR = 5.15; 95%, CI = 1.05–25.3). The chance for registration as a PBMD was twice as high among respondents who participated in volunteering, compared to those who were not volunteers (OR = 2.07; 95%, CI = 1.12–3.82). The chance for registration as a PBMD was 67% lower in the case of respondents who participated in religious practices, compared to those who did not participate (OR = 0.33; 95%, CI = 0.15–0.72). Having a relative/friend who needed and received a donation increased the chance for registration as a PBMD by 2.35 times (OR = 2.35; 95%, CI = 1.12–4.9). Living in a medium-sized town increased the chance for registration as a PBMD by a factor of 2.16, compared to those who lived in large cities (OR = 2.16; 95%, CI = 1.04–4.48). The chance for registration as a PBMD among respondents who were born in medium-sized towns was 52% lower, compared to the group of respondents born in large cities (OR = 0.48, 95%, CI = 0.25–0.93). Having a friend who registered as a PBMD increased the chance of registration as a PBMD by 3.03 times (OR = 3.03; 95%, CI = 1.88–4.9). The social activity of the mother increased the chance for registration as a PBMD by 15%, compared to respondents whose mothers were not active (OR = 1.97; 95%, CI = 1.15–3.39).

### 3.2. Step 2. Prediction of Registration as a PBMD Using the CART Method—A Visual Representation of a Decision Situation

The CART method is based on the recursive binary splitting of a set of observations. The construction of the algorithm is in the form of a series of questions, the answers to which determine the subsequent questions or finish the stage. As a result, a tree structure is obtained, which in its final nodes no longer contain questions, only answers. The principle of operation of the CART algorithm, which consists of splitting the data at the nodes, is based on one decision variable where the division is stopped when the answer to a given question does not determine the subsequent question. The CART method is characterized by recursive binary splitting, so that the parent node is always split into two child nodes. After splitting, the classification error is measured in each layer, thus ensuring a certain heterogeneity of the result category [30]. The goal of the method is the grouping and division of objects from the aspect of the examined parameters. It enables an automatic search for patterns and relationships in a comprehensive set of output data and their organization into a concise model (Figure 1). The dichotomous division of each node is aimed at the prediction or explanation of the shaping of the categorical dependent variable by the set of independent variables [31].

A well-fitted decision tree model can predict the training data set with the least misclassification cost or the highest accuracy [32]. In order to establish the prognostic determinants differentiating the respondents according to the readiness for donation, the analysis was performed by means of the CART model using *Twoing* as the splitting criterion. This method was applied considering the disproportions between categories within independent variables. All the analyzed variables were introduced into the model. In order to optimize the algorithm parameters, a training sample (60%) and a test sample (40%) were considered in the analyses. The split into training and test samples (an attempt to validate the model) showed a small discrepancy in the results (close probability), which suggests that the adopted model was accurately designed. This allowed for the correct classification in the case of 71.8% of objects, with an accuracy of classification for respondents who did not register as a PBMD as approximately 80%, whereas in the case of respondents who did register, it was approximately 50%. Table 3 presents the importance of predictors in the analyzed model.

Having a relative/acquaintance registered as a PBMD was found be the most important predictor in the model, followed by a declaration of religiosity, satisfaction with the relationship, and education level. Such variables as the respondent’s age, education of the father and his pro-social activity, having a friend who needed and received donation help (in the form of transplantation or blood transfusion), donation behaviors of the mother, place of residence, and relationships between parents exerted a smaller effect on the probability of registration as a PBMD in the examined group and did not exceed 10%. CART analysis allowed for the distinguishing of six nodes (Figure 1). The first and most important variable was having a friend who was registered as a PBMD. Among respondents who had such a person among their acquaintances and simultaneously had a secondary or lower level of education, only 13% registered as a PBMD. The probability of classification of respondents as those who did not register as a PBMD was 78%. Among respondents with higher education, the percentage of those who registered as a PBMD was higher, at 39.7%, and the probability of such a classification was 52%. Among respondents who did not have a friend who was registered as a PBMD and declared religiosity, only 10.7% registered as a PBMD. The probability of classification as a person who did not register as a PBMD was as high as 90%. The probability of not registering as a PBMD by respondents who did not have among their acquaintances a person registered as a PBMD and simultaneously did not declare religiosity was 75%.

## 4. Discussion

The presented research is among the most comprehensive studies in which a complex model was developed concerning the relative importance of the psycho-socio-demographic variables in the process of registration as a potential bone marrow donor. It provided evidence that this profile may be an important factor in making the decision about the readiness to make a donation decision of the LADS type and also confirmed that the process of registration of a donor is complex, because the ranging of variables according to importance of their effect exerted on the registration as PBMD was not always the same as that which most differentiated those who registered and those who did not. Among the factors mentioned in the hypothesis as predictors of a readiness to donate in the form of registration as PBMD, not all of them appeared to have an actual effect on making the decision.

Personal experiences. Knowing someone who declared to be a donor or someone who needed and received donation help are factors with a confirmed predictive effect [17,19,33]. This means that here, rather than an upbringing at home or examples of parents’ social activity, were of crucial importance. Such decisions were made clearly under the effect of personal experience, mainly contact with a donor (or an actual recipient). If any family factors played some role here, they were mainly friendly relationships between parents and the social activity of the mother, but these were not factors of great prognostic importance. This means that the ‘creation of a donor’s identity’ using family modelling-identification actions (e.g., by means of the model of intergenerational transmission of parental attitudes towards donation), as suggested by some studies [34], is relatively complex and therefore seems to depend on whether the person has/had acquaintances/friends showing donation behaviors or a personal contact with the environment of ill people. Here, the effect that occurred here was similar to that described by Mocan and Tekin [35], which enhanced the tendency towards donation in people who had contact with ill people (e.g., during a visit to a hospital) or were ill themselves. Therefore, the hypothesis is confirmed that it is mainly the contact with a donor that shapes the donor’s identity [19]. Similarly, the engagement in volunteering seems to pave the way for donation decisions, according to the contemporary understanding of its idea as an instrument of building good. Nevertheless, remaining in a relationship did not exert a predictive effect on the readiness to donate, as was signaled in the research [13,17,36], and this was a satisfying intimate relationship, indicated by CART as more frequent in PBMD groups, that played a prognostic role concerning the decision.

Declaration of religiosity and religious practices/religious objections. The effect of religiosity on attitudes towards transplantation was confirmed [37]. Religious beliefs seem to exert a suppressing effect on the decisions concerning donation. The declaration of religiosity was identified as a factor related with negative perception of organ donation. These results are consistent with previous findings, suggesting the effect of the declared religiosity on the probability of making a decision about potential donation, which means that the religious aspect may block donation decisions [13,15,34,37,38,39,40,41,42] as in other types of donations [12,14,43,44,45], although there are some studies which deny this [46,47]. However, it is noteworthy that the applied analyses were not completely consistent—LR suggests a suppressing effect of participation in religious practices, whereas CART emphasizes the role of religiosity. Thus, religiosity (attitude) and active religious practices (behavioral aspect) are separate categories that are worth more comprehensive investigations in the future.

Social status. People with a higher level of education seem to be more willing to make decisions about donations. The studies to date have demonstrated that PBMDs are better educated and have a more positive attitude towards science than non-donors [13,15], which seems to be a universal pattern, active irrespective of the cultural circle in which the studies are conducted and the type of donation [12,14,44,48,49,50,51]. Here, no effect was observed of a higher material status as a predictor of the decision, mentioned by other researchers with respect to, for example, blood donation or a post-mortem donation [44]. The places of birth and residence were also related to the decisions about making a donation. More frequently, the decisions about donations were made by respondents who live in towns (although the fact of coming from this environment was not the predictor of a donation decision). It is an interesting fact worthy of further studies that these were not the largest cities (usually considered as the centers of cultural or social activity) that released the readiness for donation.

State of health. A negative evaluation of the state of one’s own health appeared to be the factor suppressing readiness for BM donation. This confirms the results of other studies [16], also in relation to organ donation [52]. It is worth noting that diseases reported by the respondents (Table 1) are not always an absolute exclusion criterion, which is emphasized in the literature [53]. Thus, the importance of the education of potential donors should be emphasized concerning the criteria of exclusion from BM donation [16,54].

Gender and age. Age did not exert an effect on the readiness for donating, which differed from the results obtained by other researchers [13,15]. Moreover, no significant differences in the readiness for donation were observed according to gender. Nevertheless, women appeared to be more interested in participating in the study concerning the problem of donation [18], which does not mean, however, that they expressed a greater readiness for donation. This confirms that gender acts differently in the various groups of respondents and the types of donations [16,18,45,55,56,57,58], which requires further studies. Respondents aged 40–50 years constituted a slight percentage of the study group (13 people, less than 2.5% of the study group), while they may still be bone marrow donors.

Limitations. The study has certain limitations inherent mainly in the method of recruitment to the research group (selection) and in the method of analysis. A limitation may be the voluntary character of the group. Studies on volunteers require caution while generalizing the results, as there may exist differences between those who decided to participate in the study and those who did not [27]. It should be remembered that a sample consisting of volunteers may not be entirely representative. Volunteers differ from the rest of the population in that they reported to the study themselves. This may be associated with differences regarding other characteristics which contributed to the fact that the volunteers wanted to participate in the study (for example an interest in the problem, high level of pro-social behavior, etc.). Due to the fact that they differ from the rest of the population, the results obtained cannot always be generalized to the rest of the population, because it may be to some extent biased. Another selection limitation of this study is its geographical context, in which it was conducted. In order to verify the results, they should be replicated in other contexts (increasing external validity can be achieved by replicating studies in other populations). A limitation is also the selection by the Internet because the population of Internet users differs from the populations of individual countries (age, education, place of residence, etc.). This means that relatively rarely may the results of the conducted study be generalized to the population of the whole country or all Internet users [59]. However, very frequently, in the focus of interest of a researcher, are communities other than the ‘whole population’. Many studies concentrate rather on selected groups of individuals, which also concerns the present study.

The second limitation is inherent in the method of analysis. It should be remembered that the last stages of construction of the system is the implementation of algorithm in real-time conditions and the necessity for constant monitoring and maintenance of a set of ageing models (nearly each stochastic predictive model is subject to degradation, as the objects which it describes change), which requires the continuation of studies in the future. Thus, the models should be updated appropriately to suit the environment in the application stage [60]. Finally, although this study explained a significant proportion of the variance in ambivalence, the unexplained variance indicates that the other contributors to ambivalence should be explored.

## 5. Conclusions

In order to satisfy the demand for bone marrow transplantations and provide their appropriate and constant supply to the recipients in need, it is necessary to understand the motivating factors and barriers to donation in order to formulate and implement personalized and effective donor recruitment programs. In the presented study, factors were investigated that exert an effect on the decisions about potential bone marrow donation to unrelated people.

The specific characteristics of a potential donor were emphasized on which recruitment agencies may focus in order to constantly expand the potential bone marrow donor registries. The study, the analysis of the results and the obtained optimum model allow for the presentation of summarizing conclusions.

Stronger predictors. The performed LR and CART analyses suggest that among the factors that exert an effect on making the decision about the readiness for possible registration as a donor of the LADS type:(1)actual personal experience should be considered as an especially important stimulator (i.e., having a relative/friend who decided to register as a PBMD);(2)education (mainly on a higher level) played a predictive role in making the decision;(3)the applied methods of analysis indicate religious aspects; therefore, a declaration of religiosity or participation in religious practices as the main destimulators of the decision concerning the intention to donate. A negative assessment of the state of health also played a blocking role.

Smaller effects. Such variables as the respondent’s age, education of the father and his pro-social activity, donation behaviors of the mother, place of residence and relationships between the parents, having a relative/acquaintance who needed and received donation help (in the form of transplantation or blood transfusion) exerted the smallest effect on the probability of registration as PBMD. It is noteworthy that the above-mentioned family characteristics were especially signaled for the first time as vectors of effect (because, to date, attention has been paid mainly to the family transmission of attitudes and the value of open family communication concerning donation), which are worth detailed studies in this respect.

Practical implications for psycho-oncology. The study, analysis of the results, and the obtained optimum model allow for the presentation of practical implications:(1)Seeking potential donors in light of the obtained results. The most effective advocates of the idea of LADS in the version of registration as PBMD are individuals who are already registered, actual donors, and/or convalescents, and who survived due to a donation. Therefore, it seems that one should focus on their conscious activity in spreading the idea. They seem to be a target group in the popularization of the donation idea as leaders in the programs. For this purpose, the creation of a community of personally engaged spokesmen is substantially important (e.g., actual donors and saved recipients) who can be the spokespeople of organ donation. Those who are registered as organ donors should more often serve (be used) as supporters of organ donation and help increase the awareness and knowledge of others.(2)Are the indicated destimulators of the decision for a rational cause of the constant auto-exclusion of certain people from the groups of potential donors? It is worth spreading knowledge concerning the actual health contraindications related to donation because the common knowledge of this problem may not be entirely accurate, whereas false beliefs (of the type: ‘my allergy excludes me as a donor’) may be socially costly ignorance.(3)Slightly older people (aged approx. 40–50), of whom so few decided to take part in the research, with a primary and secondary education level, or rural inhabitants, seem to be the groups for potential interest in popularization and recruitment actions (in order to increase their awareness of their capability for donating).(4)Religious leaders may need greater clarity concerning the positive attitude of the Christian faith towards the issue of organ donation and a greater activity related to informing followers that donation does not stand in opposition to church teaching and the reduction of the potential taboo. Dispelling myths related to religion may also possibly lead to increased support for donation and a larger number of registered organ donors.(5)The proportions of the group suggest that males also may be a group of special demand for the popularization of the idea of bone marrow donation.

The proper evaluation of the characteristics of a potential donor may provide information on who to contact in order to obtain declarations of readiness for donation. Conscious and careful personalization of offers seem to favor the optimization of donation decisions (registration as PBMD). Recognition of the mechanism of donation decisions (based on a knowledge of the characteristics of the people who make such decisions) may be helpful in ensuring the more accurate identification of the groups from which potential donors recruit, where machine learning is an interesting methodological basis.

## Figures and Tables

**Figure 1 ijerph-20-05993-f001:**
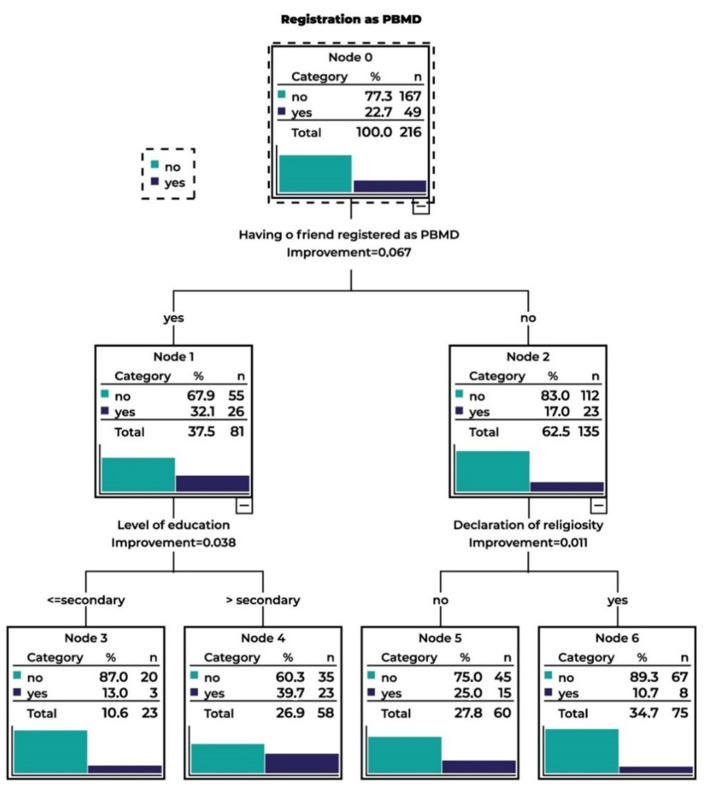
CART for the test sample.

**Table 1 ijerph-20-05993-t001:** Descriptive statistics of the study group (with data codes).

Variable	PBMD	No-PBMD		Variable	PBMD	No-PBMD
	*n* (%)	*n* (%)			*n* (%)	*n* (%)
Gender				Remaining in intimate relationship		
Females	87 (63.5)	258 (65.2)		No (2)	43 (31.4)	192 (48.5)
Males	50 (36.5)	138 (34.8)		Yes (1)	94 (68.6)	204 (51.5)
Age				Evaluation of satisfaction with the relationship		
≤40	131 (95.6)	389 (98.2)		Totally unsatisfactory (1)	2 (1.5)	17 (4.3)
>40	6 (4.4)	7 (1.8)		Moderately satisfactory (2)	26 (19.0)	75 (18.9)
Presence of chronic diseases				Totally satisfactory (3)	66 (48.2)	111 (28.0)
No (2)	132 (96.1)	375 (94.7)		Place of residence		
Yes (1)	5 (3.6)	21 (5.3)		Urban area (1)	25 (18.2)	70 (17.7)
Education level				Small town— with population over 50,000 (2)	33 (24.1)	132 (33.3)
Primary (1)	-	19 (4.8)		Mediocre town— with population over 50–500,000 (3)	38 (27.7)	79 (19.9)
Secondary school (2)	22 (16.1)	140 (35.4)		Large city— with population over 500,000 (4)	41 (29.9)	115 (29.0)
Higher/during university study (3)	115 (83.9)	237 (59.8)		Place of birth		
Assessment of material status				Rural area (1)	5 (3.6)	37 (9.3)
Poor (1)	5 (3.6)	21 (5.3)		Small town (2)	20 (14.6)	79 (19.9)
Mediocre (2)	57 (41.6)	146 (36.9)		Mediocre town (3)	37 (27.0)	127 (32.1)
Good (3)	75 (54.7)	229 (57.8)		Large city (4)	75 (54.7)	153 (38.6)
Assessment of the state of health				Education of the mother		
Poor (1)	5 (3.6)	48 (12.1)		Primary (1)	9 (6.6)	16 (4.0)
Mediocre (2)	56 (40.9)	199 (50.3)		Secondary school (2)	78 (56.9)	222 (56.1)
Good (3)	76 (55.5)	149 (37.6)		Higher (3)	50 (36.5)	156 (39.4)
Structure of family of origin				Education of the father		
Complete— two parents (1)	103 (75.2)	297 (75.0)		Primary (1)	17 (12.4)	29 (7.3)
Incomplete—broken, Single parent (2)	31 (22.6)	84 (21.2)		Secondary school (2)	61 (44.5)	177 (44.7)
Reconstructed (3)	3 (2.2)	14 (3.5)		Higher (3)	53 (38.7)	164 (41.4)
Orphanhood/lack of family (4)	-	1 (0.3)		Donation behaviours of the mother		
Siblings				No (2)	121 (88.3)	365 (92.2)
No (2)	28 (20.4)	84 (21.2)		Yes (1)	16 (11.7)	31 (7.8)
Yes (1)	109 (79.6)	312 (78.8)		Donation behaviours of the father		
Relations between parents				No (2)	118 (86.1)	348 (87.9)
Good/friendly (1)	15 (10.9)	54 (13.6)		Yes (1)	13 (9.5)	28 (7.1)
Normal (2)	58 (42.3)	187 (47.2)		Having a relative/acquaintance registered as PBMD		
Bad/conflicting (3)	64 (46.7)	155 (39.1)		No (2)	50 (36.5)	256 (64.6)
Volunteering				Yes (1)	87 (63.5)	140 (35.4)
No (2)	86 (62.8)	288 (72.7)		Pro-social activity of the mother		
Yes (1)	51 (37.2)	108 (27.3)		No (2)	96 (70.1)	333 (84.1)
Declaration of religiosity				Yes (1)	41 (29.9)	63 (15.9)
No (2)	88 (64.2)	152 (38.4)		Pro-social activity of the father		
Yes (1)	49 (35.8)	244 (61.6)		No (1)	113 (82.5)	342 (86.4)
Participation in religious practices				Yes (2)	24 (17.5)	40 (10.1)
No (2)	126 (92.0)	290 (73.2)		Own pro-social activity		
Yes (1)	11 (8.0)	106 (26.8)		No (2)	116 (84.7)	327 (82.6)
Having a relative/acquaintance who needed and received donation help				Yes (1)	21 (15.3)	69 (17.4)
No (2)	117 (85.4)	360 (90.9)				
Yes (1)	20 (14.6)	36 (9.1)				
Chronic diseases						
	Renal diseases	Allergies	Skin diseases	Cardiovascular diseases	Diabetes	Lack of data
PBMD	1 (1.3)	1 (1.3)	1 (1.3)	-	1 (1.3)	1 (1.3)
No-PBMD	2 (7.9)	6 (23.7)	6 (23.7)	3 (11.7)	2 (7.9)	2 (7.9)

**Table 2 ijerph-20-05993-t002:** Values of logistic regression coefficients.

Predictors	B	SE	Z	df	*p*	Exp(B)	95% CI for EXP(B)	
							LL	UL
Education level			9.68	2	0.008			
Education level (1)	−19.45	9501.35	0.00	1	0.998	0.00	0.00	0.00
Education level (2)	−1.09	0.35	9.68	1	0.002	0.34	0.17	0.67
Assessment of the state of health			8.04	2	0.018			
Assessment of the state of health (1)	−2.48	0.94	6.98	1	0.008	0.08	0.01	0.53
Assessment of the state of health (2)	−0.87	0.50	3.02	1	0.082	0.42	0.16	1.12
Relationships between parents			4.89	2	0.087			
Relationship between parents (1)	1.64	0.81	4.07	1	0.044	5.15	1.05	25.3
Relationship between parents (2)	0.71	0.47	2.27	1	0.132	2.04	0.81	5.16
Volunteering (1)	0.73	0.31	5.38	1	0.020	2.07	1.12	3.82
Declaration of religiosity (1)	−0.47	0.27	3.13	1	0.077	0.63	0.37	1.05
Participation in religious practices (1)	−1.12	041	7.58	1	0.006	0.33	0.15	0.72
Having a relative/friend who needed and received donation help (1)	0.85	0.38	5.19	1	0.023	2.35	1.13	4.9
Place of residence			6.63	3	0.085			
Place of residence (1)	0.29	0.38	0.58	1	0.445	1.33	0.64	2.79
Place of residence (2)	−0.10	0.36	0.07	1	0.788	0.91	0.45	1.85
Place of residence (3)	0.77	0.37	4.27	1	0.039	2.16	1.04	4.48
Place of birth			6.25	3	0.100			
Place of birth (1)	−0.91	0.57	2.59	1	0.107	0.40	0.13	1.22
Place of birth (2)	−0.37	0.38	0.94	1	0.332	0.69	0.33	1.46
Place of birth (3)	−0.73	0.33	4.76	1	0.029	0.48	0.25	0.93
Having a relative/acquaintance registered as PBMD (1)	1.11	0.24	20.60	1	<0.001	3.03	1.88	4.9
Pro-social activity of the mother (1)	0.68	0.28	6.09	1	0.014	1.97	1.15	3.39
Own pro-social activity (1)	−0.73	0.40	3.36	1	0.067	0.48	0.22	1.05
Constant	−1.04	0.31	11.26	1	0.001	0.35		

**Table 3 ijerph-20-05993-t003:** Hierarchy of importance of independent variables in the CART structure algorithm.

Independent Variable	Importance	Normalized Importance
Having a relative/acquaintance registered as a PBMD	0.067	100%
Declaration of religiosity	0.065	97.2%
Evaluation of satisfaction with the relationship	0.044	66.3%
Education level	0.038	55.9%
Place of birth	0.029	43.3%
Pro-social activity of the mother	0.024	36.1%
Assessment of material standard	0.019	27.8%
Donation behaviours of the father	0.016	23.8%
Assessment of the state of health	0.011	16.1%
Education of the mother	0.009	13.5%
Participation in religious practices	0.008	11.4%
Pro-social activity of the father	0.005	7.2%
Age	0.005	6.7%
Education of the father	0.003	5.2%
Having a relative/acquaintance who needed and received donation help	0.003	4.3%
Donation behaviours of the mother	0.002	2.7%
Place of residence	<0.001	0.3%
Relationship between parents	<0.001	0.0%

## Data Availability

The datasets analyzed during the current study are available from the corresponding author upon reasonable request.

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
