# Peer review of "‘Be the Match’. Predictors of Decisions Concerning Registration as a Potential Bone Marrow Donor—A Psycho-Socio-Demographic Study"

_ijerph, 2023, doi:10.3390/ijerph20115993_

Round 1
Reviewer 1 Report
The authors have made a good attempt to understand the factors required to become a potential bone marrow donor. It is a very good attempt at a very important issue. I have the following observation to make the research more readable and sound:
1. Research gaps should be clearly identified.
2. In the methodology section, the need for integrating regression and CART for data analysis must be justified.
3. A detailed description of the CART principle is required for the reader to better understand the CART in the present application.
4. The author should have a separate section for study implications( academic and practical).
Author Response
|
We would like to thank our Reviewers for a very accurate and constructive comments. |
|
|
Review 1 |
|
|
1. Research gaps should be clearly identified. |
There are many analyzes of donor decision-making factors, but there are no analyzes that take them into account together. This is a gap in the literature, and we added this point in the paper. |
|
2. In the methodology section, the need for integrating regression and CART for data analysis must be justified. |
The justification for the selection of the applied techniques of analysis were: 1) lack of theoretical data enabling the construction of a reliable model of the effect of the investigated variables, subjected to verification. In such a situation, the exploration-predictive model occurred to be the optimum choice, which is simultaneously a descriptive model allowing the description and presentation of the patterns in the examined population group; 2) complementarity of LR and CART – regression models focus on variables of a relatively high statistical significance, whereas decision trees do not optimize the adjustment of the model to the data, but in a sequential way divide the examined population into subgroups, based on the best prognostic variable. This allows identification of subgroups in the population predestined to the ocurrence of a specified event (TuszyÅ„ska-Bogucka, 2019). The combination of methods used is recommended as complementary and methodologically justified in medical research. It improves the quality of data analysis and is used for deeper reflection and formulating more complete conclusions. We included such an explanation, supported by literature, in our work. |
|
3. A detailed description of the CART principle is required for the reader to better understand the CART in the present application. |
A brief description of the CART method has been made. A link to an earlier work has been added, which very precisely characterizes CART as a method of researching donation decisions: „CART analysis is a nonparametric decision tree methodology that has the ability to efficiently segment populations into meaningful subgroups. Described as flexible and easy to interpret, CART can supplement traditional analysis to analyse patterns of prosocial behaviours at an individual level, even for conditions with a low prevalence. Such a decision was made because the CART methods are a suitable alternative in the explanation of potentially complex interactions. The CART utilizes comprehensive computerized search and sorting techniques in order to identify useful structures of the tree for classification of data from several groups. In the case of the CRT technique, the classification of the variables classification potential is assessed with relation to the division of threshold value. A single best predictor (the one the optimum cut-off point of which maximizes the number of correct classifications among diagnostic categories) is selected as an initial variable on the top of the hierarchical tree. Objects with values lower than the cut-off point are transferred to one category, while those with values higher than the cut-off point are transferred to the other area of the hierarchical tree. The cut-off points are subsequently assessed step by step for the remaining predictors. A classification tree is generated which grows until the maximum classification is achieved, or further division is considered as cost ineffective, which means that the CART procedure provides a ‘maximum tree’ in which a maximum division of the diagnostic groups is achieved. The maximum tree is subsequently cut to an ‘optimum tree’, which is the best tree based on the accuracy of prognosticating. In other words, the variables are switched off when the ‘cost’ of addition of additional variables is high with respect to the number of additional correct classifications. Using the iterative algorithm, the respondents were qualified into increasingly more homogenous subgroups, with similar changes in the cognitive assessment profiles, allowing a more nuanced interpretation of the effect of various factors on the decision (Knable et al. 2002; Li and Rapkin 2009). It is interesting to see how the predictive power of these 2 methods (LR and CART) differ on a dataset. CART was implemented at their best performance on the dataset and applied on the test set and compared to a logistic model. The CART is presented graphically; the root node (undivided data) first branches into 2 descendent nodes according to the independent variables. Within each branch, the descending tree continues assessing the remaining independent variables to determine which variable results in the best split. This recursive partitioning continues until a termination criterion is reached. At the point where no further split can be made, a terminal node is established (Fig. 1). Gini impurity function was calculated as the measure of node ‘purity’” (TuszyÅ„ska-Bogucka, 2019). |
|
4. The author should have a separate section for study implications( academic and practical). |
In our work, we have included suggestions for practical educational and recruitment activities. In the revised version, these are clearly marked as "practical implications", as suggested by the Reviewer. |
Reviewer 2 Report
The first sentence in the abstract: simplify to (for example) The study aimed to understand factors that support a decision to become a potential bone marrow donor....
Page 1 line 12 - Data - the word data is plural - should be Data were collected using......
Page 1 line 21 on - very wordy - needs to be simplified
other copy edits needed in abstract
Overall - introduction needs to more clearly walk through the background of the study and lead to the purpose of the study very clearly stated. I find the narrative needs to be reviewed, reduced duplicated content and sentences. Right now it is very wordy and doesn't clearly go from broad overview to more detailed focus about what is being studied. Some specific examples:
Page 1, line 31 - introduction -might start even more generally to bring reader into the paper
-"Despite its....is repeated in line 42. I find that sentence difficult to understand
Line 44 - Multiplicity sentence is also repeated
line 48 an instead of and
quote on page 53 - where does this come from??
line 35 - reference style - is not standard - needs to be fixed
Methods - needs to be thoroughly reviewed and checked for grammar.
For example - what does the first sentence on line 95 mean? "was applied" to what??
Data - plural - Data were collected.....
Participant recruitment - I am confused about what is meant by a social media website (2020 - 2021). Was it social media or on a website or both? Can you say which social media platform? This will also be related to limitations - who was following this social media account, etc.
It says the duration of the study was not limited - well the survey was open for a certain period of time - what is the time that it was open- Jan 2020 to Dec 2021?? or what?
There is no detailed discussion about the questionnaire - where did the questions come from? ARe they all custom questions developed by author or were they standard questions based on what was discerned from the literature review was important?
Need to include references for the method of analysis, SPSS, etc.
I am not a biostatistician so not sure if the analysis is reported in the best way. I do like Table 2. I prefer left justified text for tables - find it easier to read.
Figure 1 is too small and needs to be adjusted. 34.7 instead of 34,7
Discussion - there is some duplication - feel that the discussion could be made a little more succinct
Line 266 - is it declaration of religiosity or might it be Reports???
Limitations - who are authors trying to generalize the results to? Why does this contribute to the audience?
first sentence of limitations is very awkward - "voluntary character" of the group??
Review the Implications again and try to simplify.
The entire paper needs to be reviewed again with the aim to walk the reader through the story from start to finish. I believe the report of findings (and education for example could be reported once rather than as a most important finding and then again later in the findings) needs to be more succinct. I feel that the discussion was clearer than other sections, but could be shortened likely or duplication removed. Implications can be reviewed and edited for clarity.
Author Response
|
We would like to thank our Reviewers for a very accurate and constructive comments. |
|
|
Review 2 |
|
|
Page 1, line 31 - introduction -might start even more generally to bring reader into the paper |
The introduction has been extended, we hope that it will better introduce the reader to the subject of the work. |
|
-"Despite its....is repeated in line 42. I find that sentence difficult to understand |
Inaccurate phrase has been removed. |
|
line 48 an instead of and quote on page 53 - where does this come from?? |
Inaccurate phrase has been removed. |
|
line 35 - reference style - is not standard - needs to be fixed |
The reference has been corrected acording to IJERPH standard. |
|
Methods - needs to be thoroughly reviewed and checked for grammar. |
The proof was made. |
|
For example - what does the first sentence on line 95 mean? "was applied" to what?? |
Inaccurate wording has been removed. |
|
Data - plural - Data were collected..... |
Inaccurate wording has been corrected. |
|
Participant recruitment - I am confused about what is meant by a social media website (2020 - 2021). Was it social media or on a website or both? Can you say which social media platform? This will also be related to limitations - who was following this social media account, etc. |
The information has been clarified. |
|
It says the duration of the study was not limited - well the survey was open for a certain period of time - what is the time that it was open- Jan 2020 to Dec 2021?? or what? |
The information has been clarified. |
|
There is no detailed discussion about the questionnaire - where did the questions come from? ARe they all custom questions developed by author or were they standard questions based on what was discerned from the literature review was important? |
The information has been clarified. |
|
Need to include references for the method of analysis, SPSS, etc. |
The information has been clarified. |
|
I am not a biostatistician so not sure if the analysis is reported in the best way. I do like Table 2. I prefer left justified text for tables - find it easier to read. |
The table has been reformatted. |
|
Figure 1 is too small and needs to be adjusted. 34.7 instead of 34,7 |
The numbers have been corrected. The figure was resized (x10). |
|
Discussion - there is some duplication - feel that the discussion could be made a little more succinct |
The discussion has been corrected. |
|
Line 266 - is it declaration of religiosity or might it be Reports??? |
As indicated in the text, it is a declaration of religiosity and participation in religious practices (cult). |
|
Limitations - who are authors trying to generalize the results to? Why does this contribute to the audience? |
The information has been clarified. |
|
first sentence of limitations is very awkward - "voluntary character" of the group?? |
It should be remembered that a sample consisting of volunteers may not be entirely representative. Volunteers differ from the rest of the population in that they reported to the study themselves. This may be associated with differences regarding other characteristics which contributed to the fact that the volunteers wanted to participate in the study (for example an interest in the problem, high level of pro-social behaviour, etc.). Due to the fact that they differ from the rest of the population the results obtained cannot always be generalized to the rest of the population, because may be to some extent biased. |
|
Review the Implications again and try to simplify. |
The Implications have been corrected. |
|
The entire paper needs to be reviewed again with the aim to walk the reader through the story from start to finish. I believe the report of findings (and education for example could be reported once rather than as a most important finding and then again later in the findings) needs to be more succinct. I feel that the discussion was clearer than other sections, but could be shortened likely or duplication removed. Implications can be reviewed and edited for clarity. |
The text was reconstructed (mainly discussion and conclusions, necessary information added). We hope this has increased the readability of the text. |
Reviewer 3 Report
I read the manuscript entitled 'Be The Match'. Predictors of Decision About Registration as a Potential Bone Marrow Donor. Psycho-Socio-Demographic Study Using Machine Learning Methods". The subject is interesting, although the results of the study seem commonplace. Here are my comments
- The title needs improvement the phrase "Be The Match" (has it related to NMDP?) I think it should be deleted. I would suggest a title like “Predictors of Decision About Registration as A Potential Bone Marrow Donor. A psycho-socio-demographic study”
-A qualitative study must include sample power analysis. Certainly, the authors should report how they calculated the sample size.
-It would be helpful if the authors mentioned the exact time frame in which the study was conducted.
- Table 1 needs improvement, consider whether it should be split, or moved to an appendix.
- This is a study using a convenience sample, and this is a problem in generalizing results, so I would have expected the authors to mention this in the study's limitations.
Author Response
|
We would like to thank our Reviewers for a very accurate and constructive comments. |
|
|
Review 3 |
|
|
The title needs improvement the phrase "Be The Match" (has it related to NMDP?) I think it should be deleted. I would suggest a title like “Predictors of Decision About Registration as A Potential Bone Marrow Donor. A psycho-socio-demographic study” |
We would like to keep the phrase 'Be the Match' in the title, as we use it in a series of works on the psychology of donation (f.e. https://link.springer.com/article/10.1007/s12144-019-00319-5#Fn9). Prosimy o akceptacjÄ™ tytuÅ‚u: ‘Be the Match’. Predictors of Decision About Registration as A Potential Bone Marrow Donor. A psycho-socio-demographic study.
|
|
A qualitative study must include sample power analysis. Certainly, the authors should report how they calculated the sample size. |
Logistic regression – sample size The calculator for Logistic regression, UCSF Clinical and Traslational Science Institute, calculator https://sample-size.net/logistic-regression-sample-size/ Sample size required to to compare an odds ratio from logistic regression to 1. The model is of a continuous explanatory variable and a binary outcome variable. The OR is for a one-standard-deviation (SD) increase in the continuous explanatory variable. Parameters: Odds ratio =2.00 Proportion of sample in group 1 =0.10 Proportion of sample in group 2 = 0.90 Standard deviation =1 α (two-tailed) = 0.05 Threshold probability for rejecting the null hypothesis. Type I error rate. β = 0.20 Probability of failing to reject the null hypothesis under the alternative hypothesis. Type II error rate. Power 1-β = 0.80 Sample size for single independent variable: n1(Raw) = Raw calculation (i.e., without VIF) for size of group 1 = 18.3459. The calculator seeks a value of n1 such that the equations below will yield a probability of tα (given DF and NCP) that is equal to the value of β you selected above. n0(Raw) = Raw size of group 0 = (q0/q1) * n1(Raw) = 165.1130. DF = Degrees of freedom = n1(Raw) +n0(Raw) - 2 = 181.4589 tα = Inverse of the two-tailed T distribution given probability of 1-(α/2) and DF of 181.4589 = 1.9731 NCP = Non-Centrality Parameter = ln(OR)/√1/n1(Raw) + 1/n0(Raw) = 2.8165 Probabilty from the non-central t distribution of tα, given DF and NCP above = 0.199994. (If n1(Raw) , this should closely approach the value of β you selected above.) n1(Raw) and n0(Raw) are rounded up to next highest integer.
Results: n1(Raw): 19 n0(Raw): 166 nTotal(Raw): 185
2. Calculation using Variance Inflation Factor: Additional parametr: VIF ρ2 = 0.5 VIF = 1/(1-ρ2) = 2 Results: n1(VIF) = round-up(n1(Raw) * VIF) = 37 n0(VIF) = round-up(n0(Raw) * VIF) = 331 nTotal(VIF) = n1(VIF) + n0(VIF) = 368
Reference: Chow S-C, Shao J, Wang H. Sample size calculations in clinical research. 2nd ed. Boca Raton: Chapman & Hall/CRC; 2008. Section 3.2.1, page 58. Hsieh FY, Bloch DA, Larsen MD. A simple method of sample size calculation for linear and logistic regression. Stat Med. 1998;17(14):1623-34.
|
|
It would be helpful if the authors mentioned the exact time frame in which the study was conducted. |
The information has been clarified. |
|
Table 1 needs improvement, consider whether it should be split, or moved to an appendix. |
The table has been reformatted. We leave it to the editor's decision to transfer it to the appendix. |
|
This is a study using a convenience sample, and this is a problem in generalizing results, so I would have expected the authors to mention this in the study's limitations. |
The information has been clarified. |